# Improving Multi-Tumor Biomarker Health Check-Up Tests with Machine Learning Algorithms

**DOI:** 10.3390/cancers12061442

**Published:** 2020-06-01

**Authors:** Hsin-Yao Wang, Chun-Hsien Chen, Steve Shi, Chia-Ru Chung, Ying-Hao Wen, Min-Hsien Wu, Michael S. Lebowitz, Jiming Zhou, Jang-Jih Lu

**Affiliations:** 1Department of Laboratory Medicine, Chang Gung Memorial Hospital at Linkou, Taoyuan City 33305, Taiwan; mdhsinyaowang@gmail.com (H.-Y.W.); cchen@mail.cgu.edu.tw (C.-H.C.); b9209011@cloud.cgmh.org.tw (Y.-H.W.); 220/20 GeneSystems, Inc., Rockville, MD 20850, USA; peichang@2020gene.com (S.S.); mlebowitz@2020gene.com (M.S.L.); 3Program in Biomedical Engineering, Chang Gung University, Taoyuan City 33301, Taiwan; 4Department of Information Management, Chang Gung University, Taoyuan City 33301, Taiwan; 5Department of Computer Science and Information Engineering, National Central University, Taoyuan City 32001, Taiwan; jjrchris@g.ncu.edu.tw; 6Graduate Institute of Biomedical Engineering, Chang Gung University, Taoyuan City 33301, Taiwan; mhwu@mail.cgu.edu.tw; 7Department of Medical Biotechnology and Laboratory Science, Chang Gung University, Taoyuan City 33301, Taiwan; 8Department of Medicine, College of Medicine, Chang Gung University, Taoyuan City 33301, Taiwan

**Keywords:** tumor marker, machine learning, health check-up, cancer screening

## Abstract

Background: Tumor markers are used to screen tens of millions of individuals worldwide at annual health check-ups, especially in East Asia. Machine learning (ML)-based algorithms that improve the diagnostic accuracy and clinical utility of these tests can have substantial impact leading to the early diagnosis of cancer. Methods: ML-based algorithms, including a cancer screening algorithm and a secondary organ of origin algorithm, were developed and validated using a large real world dataset (RWD) from asymptomatic individuals undergoing routine cancer screening at a Taiwanese medical center between May 2001 and April 2015. External validation was performed using data from the same period from a separate medical center. The data set included tumor marker values, age, and gender from 27,938 individuals, including 342 subsequently confirmed cancer cases. Results: Separate gender-specific cancer screening algorithms were developed. For men, a logistic regression-based algorithm outperformed single-marker and other ML-based algorithms, with a mean area under the receiver operating characteristic curve (AUROC) of 0.7654 in internal and 0.8736 in external cross validation. For women, a random forest-based algorithm attained a mean AUROC of 0.6665 in internal and 0.6938 in external cross validation. The median time to cancer diagnosis (TTD) in men was 451.5, 204.5, and 28 days for the mild, moderate, and high-risk groups, respectively; for women, the median TTD was 229, 132, and 125 days for the mild, moderate, and high-risk groups. A second algorithm was developed to predict the most likely affected organ systems for at-risk individuals. The algorithm yielded 0.8120 sensitivity and 0.6490 specificity for men, and 0.8170 sensitivity and 0.6750 specificity for women. Conclusions: ML-derived algorithms, trained and validated by using a RWD, can significantly improve tumor marker-based screening for multiple types of early stage cancers, suggest the tissue of origin, and provide guidance for patient follow-up.

## 1. Introduction

Early detection of cancer is one of the major keys to improving survival of patients by enabling early treatment, including surgical removal of the localized solid tumor, before metastasis when survival rates reduce sharply to less than 50% even with state-of-the-art systemic therapies [1]. Many cancers can take years to develop to metastasis from their original lesions [2], providing the opportunity for detection of cancer at this early stage. Several tools are currently used for cancer screening [3], such as low-dose chest computed tomography (CT) for lung cancer, mammography for breast cancer, pap smear for cervical cancer, and stool occult blood for colorectal cancer. The performance of these tools is variable. This is in part due to their operator-dependent nature [4,5], limited availability, and difficulty of use. For example, pronounced disparities in the availability of low-dose chest CT instrumentation restrict its wide application [6], while the collection of specimens by non-medically trained individuals jeopardizes the accuracy of stool occult blood testing [7]. Furthermore, these tools usually detect only one cancer type, meaning that individuals may need to visit multiple medical services to receive different screening tests. These disadvantages lead to low compliance with cancer screening by these tools [8].

Serum protein tumor markers like CEA, AFP, CA-125, CA-19.9, PSA, etc., have been used for decades to aid in the diagnosis and management of a variety of cancers. Except for PSA, most international guidelines recommend routine use of these markers for monitoring cancer recurrence or therapy response, but not for screening or early detection [3]. Nevertheless, in Asia, these tumor markers are routinely measured as part of yearly physical exams for tens of millions of individuals each year and have been successfully used for the early detection of cancer [3,9]. Based on interviews with the medical directors of dozens of “health check-up” and physical examination centers in Japan, Taiwan, Korea, China, and Russia, the popularity of this testing approach appears to be growing.

Over the past 30 years tens of billions of dollars have been invested to discover and validate alternatives to serum protein tumor markers. The alternative targets include circulating tumor DNA, microRNA, and circulating tumor cells [10,11,12]; yet to date, none of these approaches has seen widespread clinical adoption, either because of cost or a lack of prospective or real-world validation. In order to accurately assess the efficacy of the tests in a real world asymptomatic population, it is absolutely crucial to use real world evidence (RWE) derived from real world data (RWD) to validate findings from case-controlled studies and to generalize the findings back to real world situations [13,14,15]. As immunological measurement of tumor markers has been performed over a number of years on a large population of individuals in a pre-diagnostic mode, RWD now exists for these biomarkers. While single tumor markers may not perform well enough, using a marker panel consisting of multiple tumor markers can significantly improve the performance of cancer screening tests [3,9,10]. Thus, tumor marker measurement is now routinely performed in Eastern Asia and has resulted in the early detection of cancers in the asymptomatic population.

Supervised machine learning (ML) is a good analytical method for solving classification problems through identification of implicit data patterns from complex data. The ML method outperforms some traditional statistical methods (i.e., univariable/multivariable analysis) because of its excellent ability to handle complex interactions between large numbers of predictors and good performance in non-linear classification problems. ML has been successfully applied in several clinical fields and outperforms traditional statistical methods [16,17,18]. 

In this study, we develop a cancer screening ML model using the largest database of RWD of individuals screened with standard tumor markers to date (from Chang Gung Memorial Hospital (CGMH), Linkou Branch), to detect implicit cancers from asymptomatic individuals accurately, rapidly, and in a ready-to-use manner based on RWE. In addition, we validate the robustness and generalization of the ML-educated algorithm through external validation using an independent testing dataset from another medical center (CGMH, Kaohsiung Branch). We also create and validate an ML-deduced algorithm to suggest the tissue of origin of the cancer after a case has been identified as suspicious for cancer.

## 2. Materials and Methods

### 2.1. Patient Eligibility

The study was approved by the Institutional Review Board of Chang Gung Medical Foundation (No. 201601798B0). We followed the Standards for Reporting of Diagnostic Accuracy 2015 [19]. Patient records were anonymized and de-identified prior to analysis. We included 27,938 (12,622 men and 15,316 women) apparently asymptomatic individuals who had at least one voluntary test with a panel of tumor markers between May 2001 and April 2015 at the Linkou or Kaohsiung branch of CGMH. All individuals had complete data on 6 tumor markers (AFP, CEA, CA19-9, CYFRA21-1, SCC, and PSA) for men and 7 tumor markers (AFP, CEA, CA19-9, CYFRA21-1, SCC, CA125, and CA15-3) for women. AFP, CEA, CA19-9, SCC, PSA, CA125, and CA15-3 were measured using commercially available kits (Abbott Diagnostics, Abbott Park, IL, USA). CYFRA21-1 was analytically determined with a commercially available kit (Roche Diagnostics Corp., Indianapolis, IN, USA). All assays met the requirements of the College of American Pathologists (CAP) Laboratory Accreditation Program, ensuring the results were accurate and reproducible. All the cases were tracked for at least 12 months subsequent to testing. Cancer diagnoses were obtained from the Taiwan Cancer Registry of the Ministry of Health and Welfare to determine whether each patient had received a new diagnosis of cancer subsequent to tumor marker testing. Of the 12,622 men, 186 received a new diagnosis of cancer. Similarly, of the 15,316 women, 156 received a new diagnosis of cancer. The data from CGMH Linkou branch, included 8415 men (124 cancers) and 10,211 women (108 cancers). In the dataset from CGMH Kaohsiung branch, 62 out of 4207 men and 52 out of 5105 women received a new diagnosis of cancer.

### 2.2. Training, Validating, and Testing Cancers Screening ML Models

We used the data from CGMH Linkou branch to train ML models and used data from CGMH Kaohsiung branch as the independent third-party testing dataset for validating both the male and female models. The input features included the tumor marker values, age, and gender. Since the sourced data is RWD, the datasets were extremely imbalanced, with the ratio of cancer cases to non-cancer cases around 1:100. Given this imbalance, we applied a random subsampling approach to build the model. It is true that classical machine learning algorithms such as support vector machine (SVM) and random forest (RF) are generally robust enough to cope with imbalanced data. However, when the data are extremely imbalanced (e.g., around 1:100 in the study), additional techniques should be adopted to improve the classification performance. For example, SVMs minimize the error over the entire dataset in order to generate these models, so they are biased towards the majority class when the imbalance is severe [20]. RF induces each constituent tree from a bootstrap sample of the training data [21]. When using an extremely imbalanced dataset, there is a significant probability that a bootstrap sample contains few or even none of the minority class, resulting in a tree with poor performance for predicting the minority class [22]. Subsampling of the majority group is a well-known technique to deal with extremely imbalanced datasets. The subsampling method is simple and not inferior to other methods in mitigating data imbalance [22]. Moreover, subsampling uses real world data and does not create artificial data like other oversampling methods.

We repeated the subsampling 200 times and internally cross validated the ML models based on the average area under the receiver operating characteristic curve (AUROC), sensitivity, and specificity. The internal cross validation was conducted by using partial data from CGMH Linkou branch to train ML models and using the other partial data from CGMH Linkou branch (unseen in the training process) to validate the ML models. ML models trained by the data from CGMH Linkou branch were validated by using the independent third-party testing dataset from CGMH Kaohsiung branch. The design is illustrated in Appendix A. ML algorithms including logistic regression (LR), random forest (RF), and support vector machine (SVM) were used. Cutoffs of ML models were determined based on the Youden index of the receiver operator characteristic (ROC) curves by the data from CGMH Linkou branch (training dataset). The cutoffs were used to classify low versus non-low risk groups. Furthermore, the non-low risk group was further equally divided into mild, moderate, and high-risk groups based on their predictive probabilities. The aim of subgrouping was to correlate the risk-stratification to the clinical prognosis but not keep these groups equal. The cutoffs were determined by the following two steps: (1) determining the cutoff (cutoff No. 1) that discriminates the low risk group from the non-low risk group; (2) based on cutoff No. 1, we determined cutoff No. 2, cutoff No. 3, and cutoff No. 4 for stratifying the mild, moderate, and high-risk groups based on their predictive probabilities. Cutoff No. 2 and No. 3 were determined intuitively by linearly dividing the probability. The cutoffs are illustrated in Appendix A. The details of tuning ML algorithms are summarized in the Appendix A.

### 2.3. Organ System for Localizing Tissue Origin

A second algorithm was developed to suggest the most likely organ system from which a cancer has originated in order to simplify the follow-up processes when an at-risk case is identified by the cancer screening algorithms. Each organ system label included several different cancer types, and all the cancer types within an organ system label are highly related (Appendix A). Moreover, all the cancer types within an organ system label are usually treated by the same medical specialist. The model used the top-N nearest-neighbor method for the analysis of tumor tissue origin based on organ systems. N was determined to be 10 in the preliminary trial. For the prediction of tissue origin, the 10 nearest cases were clustered based on the organ system label. We calculated the percentage of each organ system label in these 10 cases. The percentage of each organ system was used to rank the most likely (top N) organ systems. The second algorithm was based on the tumor marker feature space. Model performance was evaluated by repeated 5-fold cross validation for 10 times.

### 2.4. Metrics Used for Comparison of Various Models for Cancers Screening

We adopted metrics including sensitivity, specificity, accuracy, positive predictive value (PPV), ROC curve, and AUROC to access and compare the performance of the ML models.

### 2.5. Statistical Analyses

We used a Chi-squared test to analyze the distribution of cancer cases among training and independent testing datasets, and Fisher’s exact test was used for analysis when case number was less than 5. The confidence intervals for sensitivity, specificity, and accuracy were estimated using the calculation of the confidence interval for a proportion in one sample situation. Furthermore, the confidence intervals of AUROCs were determined using the nonparametric approach. We compared AUROCs by a nonparametric approach proposed by Delong et al. [23]. We followed the data analysis plan (MAQC) proposed by Shi et al. [24]. Both internal cross validation and external validation with independent testing data were used to test the robustness/reproducibility of the machine learning approach.

## 3. Results

### 3.1. General Characteristics of the Datasets

Algorithm development was performed using the datasets obtained from CGMH Linkou branch. The male dataset consisted of tumor biomarker values and age from 8415 subjects, while the female dataset consisted of tumor biomarker values and age from 10,211 subjects. To train the ML algorithms, a balanced subset of subjects ultimately diagnosed with cancer and subjects who were not diagnosed within the follow-up period was randomly selected. The biomarker values and ages of these subjects were used to train the algorithms. This subsampling process was performed 200 times (random subsampling cross-validation) and then internally validated against the entire dataset. Completely independent datasets for both males (*n* = 4207) and females (*n* = 5105) were obtained from CGMH Kaohsioung branch and used for independent validation. Mean and median biomarker values for the training and independent datasets are reported in Table 1a. The statistics in Table 1a are from one random subsampling. We also report the demographic statistics of 200 subsamplings in the Appendix A. As expected, due to the selection of balanced training data vs. the real world nature of the independent dataset, mean biomarker values vary widely, while median values are much more consistent between the datasets. Thus, median with interquartile range (IQR) is a more appropriate metric to compare groups. Based on these values, the distribution of tumor marker values in training datasets overlapped with those in the independent testing datasets. The distribution of cancer types within the training and independent datasets for both men (Table 1b), and women (Table 1c) are comparable. There are 19 cancer types in both groups. In the overall datasets, the top three cancer types for men originated from the colon, liver, and prostate while for women, the top three were breast, cervical, and thyroid.

### 3.2. Performance Characteristics of the Models

For men, ML models using a LR algorithm consistently outperformed RF and SVM models in either internal (Table 2a and Figure 1a) or external validation (Table 2a and Figure 1b) in terms of AUROC. Specific cutoff values for risk scores can be applied to define low, mild, moderate, and high-risk groups (Appendix A). The PPVs of ML models using LR algorithm were 1.99%, 2.89%, and 11.72% for mild, moderate, and high-risk groups, respectively. Test performance within the top three male cancers was comparable (Table 2b and Figure 1c). For women, ML models using RF algorithm outperformed LR and SVM (Table 2a and Figure 1d,e). Overall model performance characteristics for the female algorithms were inferior to the male algorithm, but still yielded superior performance to single marker tests (Table 2a) [3]. In the subgroup analysis, the ML models showed comparable performance in detecting the top three female cancers (Table 2b and Figure 1f). As with the male models, specific cutoff values for risk scores can be applied to the female model to define low, mild, moderate, and high-risk groups (Appendix A). The PPVs of ML models using the RF algorithm were 1.66%, 3.74%, and 10.87% for mild, moderate, and high-risk female groups, respectively. The PPVs were calculated based on the entire combined dataset (including both training datasets and testing datasets).

### 3.3. Predicting Affected Organ System

We used a second layer ML model (top-N nearest-neighbor method) to predict the possible affected organ system for individuals whose risk was mild, moderate, or high. When the top three most possible organ systems were reported, we reached a balanced sensitivity and specificity: for men, 0.8120 sensitivity and 0.6490 specificity; for women, 0.8170 sensitivity and 0.6750 specificity. The sensitivity elevated when the reported number of possible organ systems increased, but as a trade-off, the specificity decreased at the same time (Figure 2).

### 3.4. Association between the Risk Score, Cancer Stage, and Time-to-Diagnosis

Risk scores were positively correlated to the cancer stage (Figure 3). For both men and women, disease stage 0, I, and II accounted for more than half of the cases whose risk was low, mild, or moderate. By contrast, in the high-risk group, the percentage of stage I cancers decreased and the percentage of stage III increased (Figure 3). Regarding the time-to-diagnosis (TTD), we measured the time interval between tumor marker test and cancer diagnosis. For both men and women, the risk score was negatively correlated to the TTD (Figure 4). For men, the median TTD was 561, 451.5, 204.5, and 28 days for low, mild, moderate, and high-risk groups, respectively (Figure 4a). For women, the median TTD was 279, 229, 132, and 125 days for the low, mild, moderate, and high-risk groups, respectively (Figure 4b).

## 4. Discussion

Using tumor biomarkers for cancer screening/early detection is now common practice in East Asia, as tens of millions of individuals are undergoing these screening tests as part of their annual physical checkups. Using the largest reported database of RWD and external validation, we developed and validated ML-derived software that substantially improves tumor biomarker testing (Appendix A) by incorporating the values of six or more biomarkers with age and gender to assign level of risk for cancer. Further, a secondary model was developed to predict the top three most likely affected organ systems for individuals identified as at-risk. The ML algorithms are robust and help detect cancers at the earliest stages in asymptomatic individuals.

Immunoassays for tumor markers have been developed over the last several decades, and most are used for monitoring during post-therapy follow-ups but not in a pre-diagnostic mode for screening of asymptomatic individuals. Using single tumor biomarkers for cancer screening has been less robust; and even PSA, the only tumor marker widely used for cancer screening, remains controversial [25]. SCC as a stand-alone tumor marker for cancer screening has been questioned by clinical physicians including at our institution, CGMH, Taiwan [3]. Cohen et al. evaluated a number of tumor markers for cancer screening in a case-control study, and found that some markers (including AFP, CEA, CA19-9, CYFRA21-1, CA125, and CA15-3) could be used together as a panel, to improve performance [10]. Several studies have shown promising results for cancer screening (lung cancer [26,27]; multiple types of cancers [3,10]) by combining ML algorithms and tumor marker panels, because ML algorithms outperform clinical physicians in interpreting analytical results [3]. Clinicians generally interpret lab values by a “single threshold method”, which is based on pre-determined reference ranges for each individual marker. However, the reference ranges are set solely based on the value distribution within the normal population, not for cancer screening, they are not adjusted in connection with other marker levels, and do not consider age, gender or any other patient characteristic. By contrast, ML algorithms can learn and identify the specific pattern of tumor markers and clinical factors and their interdependence for discriminating cancer cases from non-cancer cases.

Recent reports have demonstrated that ML algorithms can improve diagnostic accuracy when applying multiple markers [28]. However, there are barriers that hinder wide application of ML algorithms for screening. The number one barrier is that the subjects used in clinical studies often do not mimic real world populations. Many ML algorithms are developed by case-control studies, using well defined cases to learn a decision plane capable of classifying different classes (e.g., cancer vs. non-cancer). Biomarkers of the cancerous cases used in these case control studies are analyzed after, but not before, the cancer diagnosis. Therefore, biomarkers are more representative of symptomatic patients in which tumors have already progressed, and profiles are more indicative of advanced cancers in the symptomatic patient but do not correlate to the asymptomatic population in the real world [10,26,27]. Consequently, the ML models trained by using such data should be regarded as models for cancer diagnosis but not for screening and will yield compromised results when applied to the real world asymptomatic pre-diagnostic setting. By contrast, we used the largest dataset of RWD currently available, collected over 14 years for both training and external validation of ML models to interpret tumor marker panels. These algorithms demonstrated superior performance characteristics (Table 2, Figure 1 and Figure 2), and the robustness of these models was validated by external validation. In addition, all the subjects in this study were individuals undergoing a yearly health check-up with no prior indication of cancer and thus represent a real world asymptomatic population [3,9]. Due to the use of RWD, the ML model developed herein is ready for immediate application in the real world setting.

As shown in Figure 2, we report the top three most possible (more than one) organ systems for physicians or caregivers to further investigate the possible origin of cancer. Based on the design, a subject will be labeled with top three most possible organ systems, for example: “Chest”, “Ear, Nose, and Throat”, and “Gastrointestinal” in the report. An experienced physician would further check the individual’s specific exposure history (e.g., smoking, fine particulate matter (PM_2.5_)) and arrange a low-dose chest CT because the reported pattern may imply a clinical picture of cancerous change over the pharynx/larynx and lung, which can be classified to organ system labels of “Ear, Nose, and Throat” and “Chest”, respectively. Regarding the aspect of analysis and reporting, we used all the cancer cases to cross-validate the KNN model and evaluate by what N we could help physicians identify the tissue origin of cancer accurately. The reason for reporting only the top three most possible organ systems was based on (1) achieving the best balance between sensitivity and specificity that could be achieved, and (2) a top three possible tissue origin would be reasonable and actionable for further clinical survey in most of clinical settings.

Another barrier of applying ML algorithms in clinical routine is the lack of actionable recommendations [28]. To generate a ML model usable in the clinical setting, we designed a two-layer model for cancer screening. The first layer ML model identifies the relative risk of individuals for developing cancer in the near term, and the second layer ML model predicts the top three most likely affected organ systems. In the first layer, four risk levels are reported. The PPVs for the different risk levels, together with their correlations to cancer stage at diagnosis (Figure 3) and time to diagnosis (Figure 4) suggest different follow-up procedures (Figure 5). TTD is an important factor in determining post screening patient follow-up. The short TTD (2–4 months) for individuals in the high-risk group suggests these patients should be followed up in the short term. For individuals at both moderate and high risk, based on TTD, next steps likely should include referral to a specialist for further work-up, ideally within 6 and 2 months, respectively. Advanced diagnostic tools could be used for confirmation; for example, low-dose chest CT could be used when “chest” is predicted as the most likely organ system. The action following a positive risk call should be at the discretion of physicians and depend heavily on standard health practices in different countries. In Taiwan, a colonoscopy will be used as the diagnostic tool for following up at-risk individuals because it is affordable and available in nearly every hospital. At the same time, since many cases of mild risk may be at a pre-cancer status which is a dynamic stage between developing into a local tumor and being eliminated by the immune system, and the half-life of most tumor markers is less than several days [29], we would recommend that these individuals receive repeat tumor marker screenings in one month to verify the level of risk. Future repeat testing at 6–12-month intervals to monitor changes in overall score and individual biomarker levels would seem prudent.

We tried several ML algorithms and found that simple/linear algorithms perform well in classification problems within the medical field. We also found the same phenomenon in other classification studies [16,17,18,30,31,32,33,34]. We do not completely understand the reason behind this fact but hypothesize that the phenomenon of “simple/linear algorithms perform well” in the study could be attributed to the representativeness of the features (i.e., serum protein tumor markers). All of the serum protein tumor markers used in the study were developed and validated for specific cancer types (e.g., CEA for colorectal cancer; CA19-9 for pancreatic cancer). In the history of marker development, weak prediction markers were filtered out and only strong predictor markers were selected. On this basis, the correlation between the serum protein tumor markers would not be too complicated. Thus, simple linear algorithms can perform well on the type of dataset.

To the best of our knowledge, the studies [3,9] published previously by our team are among the few studies addressing tumor markers in cancer screening. The rarity of such studies is likely attributable to the fact that long-term follow-up for a large asymptomatic cancers screening population is rare and difficult. Thus, we compared the current results with our previous publications. In 2015, we analyzed 12 years of tumor marker data and showed that a tumor marker panel could be used for cancer screening. In 2016, we demonstrated that harnessing machine learning algorithms could improve the utility of a tumor marker panel for cancers screening in an internal validation setting [3]. On this basis, in the current study we further demonstrate the reproducibility of the machine learning algorithms by testing through external validation. Currently, some commercial products in development, including CancerSEEK [10] and Grail [12], utilize nucleic acid sequencing technology to detect cancer related genes in blood to screen for multiple cancers. Both of these products claim the ability to detect multiple types of cancers and to identify their tissue of origin. The aim of these studies is quite similar to our study and the test performances are promising; however, the tests have only been validated in case-control studies in which the cancer cases are not representative of the asymptomatic cases found in a real world population. Thus, the results of these studies would be inappropriate to compare with the results of the current study.

Velocity of tumor marker or risk score change between serial tests may be helpful information in determining associated risk and further follow-up. At the same time, there is well documented information regarding interference of tumor marker immunoassays which should be incorporated into the interpretation of some results, especially for the mild risk level. For example, heterophilic antibody reactivity can cause pseudo-elevation of multiple tumor markers [35], while poor sugar control can result in pseudo-elevation of CA19-9 [36]. Biotin-based immunoassays can be affected by individuals taking high doses of nutrient supplements containing biotin [37]. Taking a well-documented patient history and applying that information can reduce these interferences.

Given the fact that 8.7 million years of life are lost each year due to early cancer deaths resulting in $94.4 billion in lost earnings [38], a cancer screening tool for which the number needed to screen (NNS) is less than 200 [3] would serve as an appropriate and affordable tool to screen cancers and prevent premature cancer death.

We would like to acknowledge several limitations in this study. First, we used cases from only two tertiary medical centers in Taiwan to train and externally validate the ML models, respectively. The generalization of the ML models was demonstrated, however, generalizability of the models to other world regions or countries is yet to be evaluated. We suggest that a clinically useful ML model should be built using a generalizable approach and locally relevant data given the variability existing in populations or analytical systems [39]. Second, we did not use a deep learning (DL) algorithm for developing our cancer screening models. In this study, only six features (i.e., tumor markers) for men and seven features for women, plus one clinical factor age, were employed. We used classical ML algorithms to build robust models and avoid overfitting. Third, while we utilized RWD for the training and validation of the algorithms, a prospective study to evaluate the premature cancer death prevention and cost savings which can be attributed to the cancer screening tool has yet to be performed. The net financial benefit of the tool is worthy of further investigation. Last, regarding the nature of a screening tool, we selected an algorithm with balanced sensitivity and specificity, which is not 100%. In fact, tests with even 99.9% specificity still generate a lot of false positives when used to screen millions of people. We propose repeated blood testing or further investigation only for the individuals with a high algorithm score and or with biomarkers trending upwards to avoid the possible harm brought by false positives.

## 5. Conclusions

We developed and validated robust ML models capable of improving early detection of cancer and localizing the tissue origin by using RWD of tumor markers. These ML-derived algorithms are appropriate cancer screening tools demonstrating high levels of accuracy, generalizability, and affordability.

## Figures and Tables

**Figure 1 cancers-12-01442-f001:**
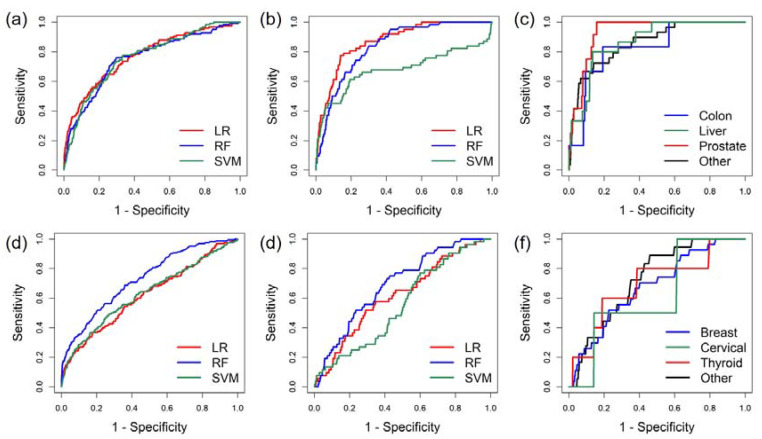
(**a**) Receiver operating characteristic (ROC) curves of internal cross validation of the cancers screening algorithms by male training dataset. LR: logistic regression; RF: random forest; SVM: support vector machine. (**b**) ROC curves of external validation of the cancers screening algorithms by male independent dataset. (**c**) ROC curves of cancers screening algorithm for male different cancer types. (**d**) ROC curves of internal cross validation of the cancers screening algorithms by female training dataset. (**e**) ROC curves of external validation of the cancers screening algorithms by female independent dataset. (**f**) ROC curves of cancers screening algorithm for female different cancer types.

**Figure 2 cancers-12-01442-f002:**
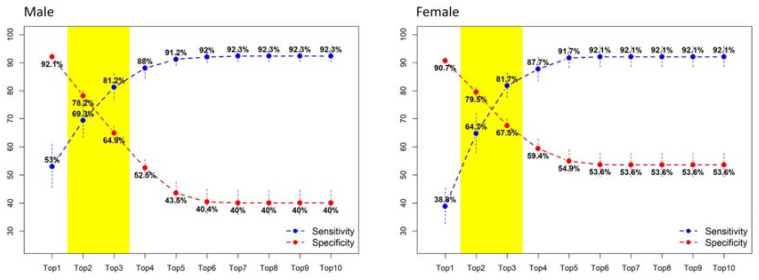
Organ system algorithm for localizing tissue origin. For the cases predicted with non-low risk scores, a k-nearest neighbor based organ system algorithm was used for localizing tissue origin. The performance of a different number of origins is evaluated. When only one possible tissue origin (i.e., top 1) is reported, the sensitivity is low, but specificity is high. In contrast, the sensitivity increases, and specificity decreases when more possible tissue origins (e.g., top 10) are reported. A balanced sensitivity and specificity of the algorithm can be achieved when the top three possible organ systems are reported.

**Figure 3 cancers-12-01442-f003:**
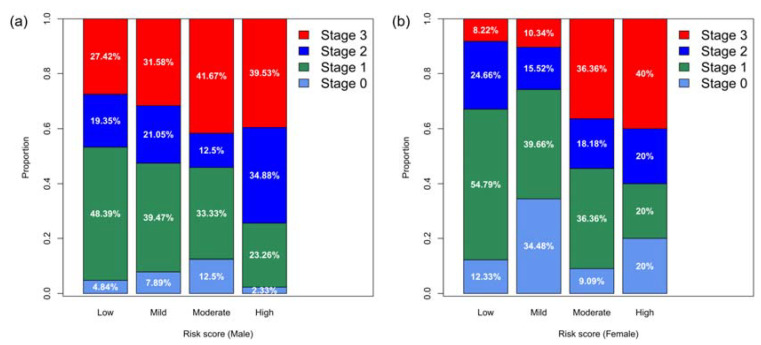
(**a**) Characteristics for different levels of risk. The distribution of stages of cancers for different levels of risk (male) is illustrated. The risk score is positively correlated to the cancer stage, namely the proportion of advanced cancer stage increases in the higher risk score categories. (**b**) Characteristics for different levels of risk. The distribution of stages of cancers for different levels of risk (female) is illustrated. The risk score is positively correlated to the cancer stage, namely the proportion of advanced cancer stage increases in the higher risk score categories.

**Figure 4 cancers-12-01442-f004:**
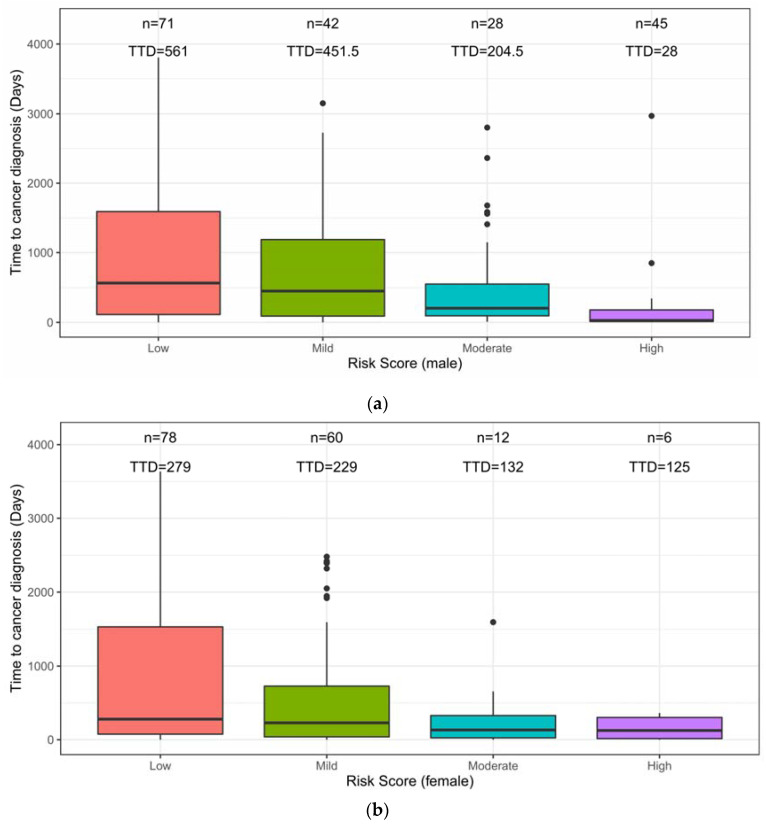
(**a**) Time to diagnosis (TTD) for cancer cases with different levels of risk (male). The median TTD is inversely correlated to the risk score: when a case is predicted with a higher risk score, a shorter TTD can be expected, thus a shorter time for additional investigation could be suggested. (**b**) Time to diagnosis (TTD) for the cancer cases with different levels of risk (female). The median TTD is inversely correlated to the risk score: when a case is predicted with a higher risk score, a shorter TTD can be expected, thus a shorter time for additional investigation could be suggested.

**Figure 5 cancers-12-01442-f005:**
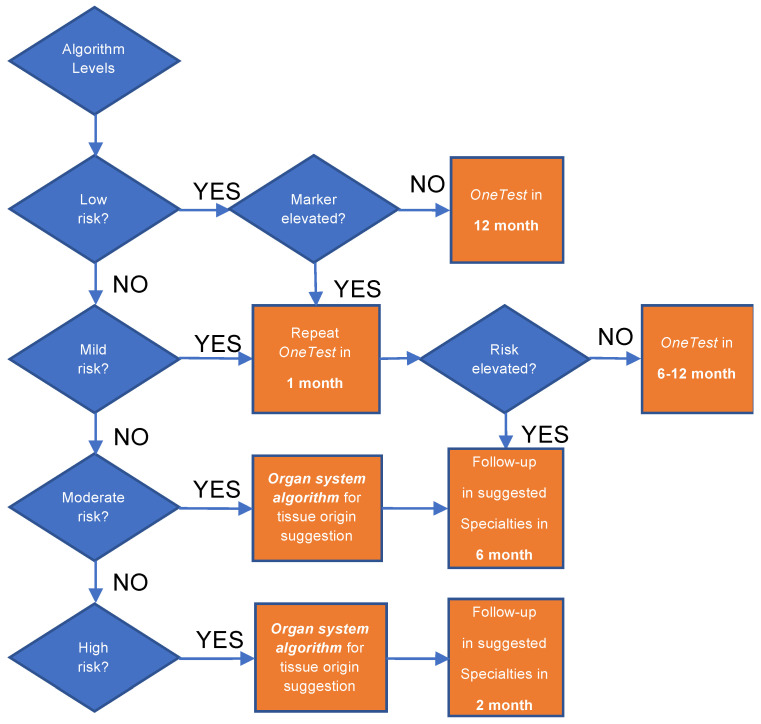
Actionable recommendation for the different levels of risk scores. When low risk is reported and all the biomarkers are below the reference ranges, 1-year follow-up is recommended. For the low risk group whose marker is elevated and for the mild risk group, 1-month repeat is recommended to exclude an elevated risk caused by interference. For the individuals whose risk score keeps elevating upon repeat and for the moderate risk group, 6-month follow-up in the suggested specialties (i.e., top 3 most likely affected organ systems provided by the Organ System Algorithm) is recommended. For the high risk group, 2-month or shorter follow-up is recommended for further investigation of cancer.

**Table 1 cancers-12-01442-t001:** **a.** Descriptive statistics of training and independent testing datasets. **b.** Cancer types in male training and independent testing datasets. **c.** Cancer types in female training and independent testing datasets.

a.								
Variation	Median	IQR	Mean	SE	Median	IQR	Mean	SE
	Male Training (*n* = 248)				Male Independent Testing (*n* = 4207)			
AFP (ng/mL)	3.40	1.99	2100.40	23,452.06	3.08	1.75	4.26	44.68
CEA (ng/mL)	2.01	1.68	3.62	10.56	1.84	1.44	2.20	1.52
CA19-9 (U/mL)	6.37	9.48	10.85	16.32	5.09	7.90	7.77	11.42
CYFRA21-1 (ng/mL)	1.61	0.98	1.86	1.15	1.35	0.91	1.54	0.83
SCC (ng/mL)	0.40	0.50	0.61	0.51	0.50	0.50	0.64	0.64
PSA (ng/mL)	1.14	1.28	8.92	103.62	0.81	0.80	1.25	2.13
Age (yr)	57.00	20.25	57.43	13.42	49.00	18.00	49.55	12.65
	**Female Training (*n* = 208)**				**Female Independent Testing (*n* = 5105)**			
AFP (ng/mL)	3.01	1.63	4.69	12.41	2.81	1.75	3.35	4.45
CEA (ng/mL)	1.42	1.18	3.83	21.21	1.27	1.02	1.52	1.07
CA19-9 (U/mL)	5.72	9.86	11.82	40.84	6.19	10.28	9.51	11.57
CYFRA21-1 (ng/mL)	1.34	1.07	1.70	1.37	1.16	0.76	1.32	0.76
SCC (ng/mL)	0.30	0.30	0.54	0.84	0.30	0.30	0.49	0.40
CA125 (U/mL)	9.80	8.19	15.29	17.63	9.39	6.84	12.13	14.00
CA153 (U/mL)	8.70	5.78	10.30	5.21	8.20	5.60	9.50	4.38
Age (yr)	49.00	14.25	50.56	10.73	47.00	16.00	47.43	11.77
**b.**								
**Cancer Type**	**Training Set**				**Independent Testing Set**			
Male	#Cancer (Total = 124)		(%)		#Cancer (Total = 62)		(%)	*p*-Value
Prostate	18		14.5		12		19.4	0.53
Liver	17		13.7		15		24.2	0.11
Colon	14		11.3		6		9.7	0.93
Lung	9		7.3		1		1.6	0.21
Pancreatic	9		7.3		7		11.3	0.52
Head and neck	8		6.5		0		0.0	0.1
Thyroid	7		5.7		0		0.0	0.13
Kidney	6		4.8		6		9.7	0.34
Leukemia	6		4.8		3		4.8	1
Gastric	6		4.8		1		1.6	0.5
Bladder	5		4.0		1		1.6	0.66
Lymphoma	5		4.0		3		4.8	1
Skin	5		4.0		2		3.2	1
Esophageal	2		1.6		3		4.8	0.42
Unknown origin	2		1.6		0		0.0	0.8
Bile ducts	2		1.6		0		0.0	1
Brain	1		0.8		1		1.6	1
Retroperitoneal	1		0.8		0		0.0	1
Testicle	1		0.8		0		0.0	1
Gastrointestinal stromal	0		0.0		1		1.6	0.72
**c.**								
**Cancer Type**	**Training Set**				**Independent Testing Set**			
Female	#Cancer (Total = 104)		(%)		#Cancer (Total = 52)		(%)	*p*-Value
Breast	31		29.8		27		51.9	0.01
Cervical	17		16.4		2		3.9	0.05
Thyroid	15		14.4		5		9.6	0.55
Colon	9		8.7		2		3.9	0.44
Lung	5		4.8		0		0.0	0.26
Liver	4		3.9		3		5.8	0.89
Ovarian	4		3.9		1		1.9	0.87
Gastric	4		3.9		1		1.9	0.87
Kidney	2		1.9		2		3.9	0.86
Leukemia	2		1.9		1		1.9	1
Skin	2		1.9		2		3.9	0.86
Unknown origin	2		1.9		0		0.0	0.8
Uterus	2		1.9		3		5.8	0.42
Bladder	1		1.0		1		1.9	1
Head and neck	1		1.0		0		0.0	1
Liposarcoma	1		1.0		0		0.0	1
Nasal neuroendocrine tumor	1		1.0		0		0.0	1
Pancreatic	1		1.0		0		0.0	1
Esophageal	0		0.0		1		1.9	0.72
Parotid	0		0.0		1		1.9	0.72

IQR: interquartile range; SE: standard error.

**Table 2 cancers-12-01442-t002:** **a.** Performance of cancers screening algorithms. **b.** Performance of screening top three types cancers.

a.				
Male	LR	RF		SVM
AUROC				
Internal CV	0.7654 (0.7596, 0.7713)	0.7555 (0.7499, 0.7612)		0.7440 (0.7380, 0.7500)
External validation	0.8736 (0.8347, 0.9125)	0.8382 (0.7984, 0.8781)		0.6804 (0.5880, 0.7728)
MCC				
Internal CV	0.0554 (0.0522, 0.0566)	0.0551 (0.0529, 0.0573)		0.0504 (0.0483, 0.0525)
External validation	0.2092 (0.1969, 0.2214)	0.1384 (0.1280, 0.1489)		0.1217 (0.1119, 0.1316)
Weighted Accuracy				
Internal CV	0.7076 (0.7032, 0.7120)	0.7311 (0.7268, 0.7354)		0.7201 (0.7157, 0.7244)
External validation	0.8171 (0.8055, 0.8288)	0.7686 (0.7558, 0.7813)		0.7098 (0.6961, 0.7235)
Sensitivity				
Internal CV	0.6604 (0.6597, 0.6611)	0.6757 (0.6749, 0.6764)		0.7028 (0.7021, 0.7035)
External validation	0.7742 (0.7616, 0.7868)	0.8387 (0.8276, 0.8498)		0.6129 (0.5982, 0.6276)
Specificity				
Internal CV	0.7418 (0.7412, 0.7425)	0.7203 (0.7197, 0.7210)		0.6996 (0.6989, 0.7003)
External validation	0.8601 (0.8496, 0.8706)	0.6984 (0.6846, 0.7123)		0.8068 (0.7948, 0.8187)
**Female**	**LR**	**RF**		**SVM**
AUROC				
Internal CV	0.6068 (0.5995, 0.6140)	0.6665 (0.6596, 0.6733)		0.5794 (0.5717, 0.5870)
External validation	0.6181 (0.5428, 0.6935)	0.6938 (0.6298, 0.7579)		0.5551 (0.4834, 0.6268)
MCC				
Internal CV	0.0218 (0.0205, 0.0231)	0.0353 (0.0337, 0.0370)		0.0272 (0.0258, 0.0287)
External validation	0.0312 (0.0296, 0.0327)	0.0562 (0.0542, 0.0582)		−0.0042 (−0.0048, −0.0036)
Weighted Accuracy				
Internal CV	0.5818 (0.5774, 0.5861)	0.6404 (0.6361, 0.6446)		0.6052 (0.6009, 0.6095)
External validation	0.5673 (0.5360, 0.5717)	0.6349 (0.6307, 0.6392)		0.4904 (0.4860, 0.4948)
Sensitivity				
Internal CV	0.4850 (0.4843, 0.4857)	0.5736 (0.5729, 0.5743)		0.3298 (0.3292, 0.3305)
External validation	0.5192 (0.5055, 0.5329)	0.6923 (0.6796, 0.7050)		0.2885 (0.2760, 0.3009)
Specificity				
Internal CV	0.6657 (0.6651, 0.6664)	0.6521 (0.6515, 0.6528)		0.8007 (0.8002, 0.8013)
External validation	0.6855 (0.6728, 0.6983)	0.6139 (0.6005, 0.6272)		0.6553 (0.6422, 0.6683)
**b.**				
**Male**	**Colon Cancer**	**Liver Cancer**	**Prostate Cancer**	**Other**
**AUROC**	0.8290 (0.6660, 0.9921)	0.8694 (0.7977, 0.9411)	0.8608 (0.7961, 0.9255)	0.9319 (0.9003, 0.9636)
**Accuracy**	0.8106 (0.7987, 0.8224)	0.8662 (0.8559, 0.8765)	0.8448 (0.8338, 0.8557)	0.8602 (0.8498, 0.8707)
**Sensitivity**	0.8333 (0.8221, 0.8446)	0.8000 (0.7879, 0.8121)	1	0.7241 (0.7106, 0.7376)
**Specificity**	0.8080 (0.7961, 0.8199)	0.8649 (0.8546, 0.8752)	0.8425 (0.8315, 0.8535)	0.8601 (0.8496, 0.8706)
**Female**	**Breast Cancer**	**Cervical Cancer**	**Thyroid Cancer**	**Other**
**AUROC**	0.6776 (0.5825, 0.7727)	0.6235 (0.1628, 1.000)	0.6940 (0.4286, 0.9594)	0.7259 (0.6363, 0.8155)
**Accuracy**	0.6453 (0.6322, 0.6584)	0.6948 (0.6821, 0.7074)	0.7444 (0.7324, 0.7563)	0.5821 (0.5686, 0.5957)
**Sensitivity**	0.5926 (0.5791, 0.6061)	0.5000 (0.4863, 0.5137)	0.6000 (0.5866, 0.6134)	0.7778 (0.7664, 0.7862)
**Specificity**	0.6456 (0.6324, 0.6587)	0.6948 (0.6822, 0.7075)	0.7445 (0.7325, 0.7565)	0.5814 (0.5679, 0.5950)

LR: logistic regression; RF: random forest; SVM: support vector machine; AUROC: area under the receiver operating characteristic curve; CV: cross-validation; MCC: Matthews correlation coefficient. The metrics are described with mean and 95% confidence interval. AUROC: area under the receiver operating characteristic curve; Other: other cancer types. The metrics are described with mean and 95% confidence interval.

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
