# Peer review of "Improving Multi-Tumor Biomarker Health Check-Up Tests with Machine Learning Algorithms"

_cancers, 2020, doi:10.3390/cancers12061442_

Round 1
Reviewer 1 Report
Paper is interesting, but there are a number of issues that need to be discussed by the authors:
- Why using subsampling? - SVM and RF should be robust enough model to cope with the data imbalance which is the standard situation for RWD
- Why not using MCC? - Matthews Correlation Coefficient is by far the best measure to summarise a classifier's performance; it is robust to imbalanced data, and provides an easier-to-read baseline to interpret
- Caveat: leave-one-out is known be biased, due to the high variance
- Lack of reproducibility: the employed methodology cannot guaranteed model reproducibility; I recommend using a more complex data analysis plan, such as those proposed by the US-FDA in the MAQC/SEQC projects
- Performances are not optimal - may an additional optimization round improve the results, that presently are good but probably not enough to support a translation to clinical practice?
- In particular, how do the results compare to other similar studies
- Final discussion is somehow too naive from a clinical point of view, providing the impression that an ad-hoc storytelling is constructed upon the results.
Author Response
Dear Reviewer,
Thank you very much for your comments on our manuscript entitled “Improving Multi-Tumor Biomarker Health Check-up Tests with Machine Learning Algorithms”. These comments are very important for improvement and further development of our study. We would like to express our sincerest gratitude to you for spending a great deal of time and effort in evaluating our manuscript. We have revised the manuscript according to the reviewers’ suggestions, taking great considerations of the reviewers’ comments and summarized our response to the comments. The locations of changes in the revised manuscript are specified to help the reviewers identify our changes. Additionally, the manuscript has been proofread by our colleagues who are native English speakers. I am submitting the revised manuscript along with a response letter indicating the changes we have made. Please do not hesitate to contact me by phone (703-867-5722), fax transmission (240-403-028) or by email (jzhou@2020gene.com) if there is any additional information that should be provided.
We look forward to receiving a favorable reply.
Sincerely,
Jiming Zhou, PhD
VP, East Asia Business Development
Director of Clinical Diagnostics
20/20 GeneSystems, Inc.
9430 Key West Avenue, Suite 100, Rockville, Maryland 20850 U.S.A.
Mobile: +1 703-867-5722
Office: 240 453-6339 x111 Fax: 240 403-028
WeChat: jz20140509
jzhou@2020gene.com www.2020gene.com

Reviewer 2 Report
The authors train and test multiple machine learning models (logistic regression, SVM, and random forest) to screen patients for many kind of cancers based on several serum biomarkers using two large datasets collected over more than a decade. Additionally, they train a KNN model to recommend organ system of origin for those patients found to be at not low-risk. The adapt their models to risk-stratify patients, resulting in physician/computer-driven actionable recommendations for additional testing, follow-ups, and specialists.
Overall, the manuscript is very interesting mostly because of the datasets. It is hard to find such a large dataset of patients (and cancer-specific biomarkers) who have been followed-up for so many years. That being said, there are some details and clarification needed that will improve the overall quality of the manuscript. Here are my comments.
- The distinction between serum tumor markers of lines 62-69 and the serum antigens of lines 70-82 seems to be that the latter has not been widely adopted due to cost or validation. But, the authors then state data is available for these serum antigens and is routinely utilized in Eastern Asia, somewhat contradicting their first statements. Are the tumor markers of lines 78-81 referring to former or latter group? Some clarification is needed.
- Further, why are these addition biomarkers mentioned in the introduction when they are not considered in the rest of the manuscript?
- 'RWD' is not defined beforehand on line 75. I assume it means real world data.
- Why repeat random subsampling only 5 times? That would cover at the most 5% of the non-cancer cases for women in the CGMH Linkou dataset and 7% of the non-cancer cases for men. That seems small.
- Can the authors further explain 'internally evaluated' on line 123?
- It is not clear how random subsampling results in "the model." One would presume that a model is cross-validated using one of the subsamplings and that some sort of consensus is built based up more random subsamplings. It is not clear if this or some other methods is the case here.
- The authors state on lines 124-125 that average statistics were computed across different subsamples. What is the variation of these performance metrics across subsamples?
- Figure 1 reports results, so it should be included in the results and not the methods section.
- How were cutoffs for non-low risk groups determined? It says based on probabilities, but given that non-low risk groups were split into three equal subgroups, it would seem it was to keep these groups equal. This is appropriate if the probabilities are uniformly distributed. What is the basis for keeping these groups equal? How are the probabilities distributed?
- Is the second algorithm based on the tumor marker feature space or the cancer/non-cancer probability space?
- How was random forest optimized?
- Are the numbers reported for the training sets in Table 1a for one random subsample? It says n=248 and n=208 for men and women respectively, which happens to be twice the number of the smaller class, so I suspect it is so. It would be appropriate to report the mean and variation of these statistics across subsamples.
- Table 1 AFP mean/SE for male training seem impossible without some clerical error.
- How is the p-value being computed in Table 1b?
- It is clear how AUCs are compared for different models and how a higher AUC would suggest higher performance of a model. How does one compare ROCs as in lines 213-215?
- I’m a bit confused how the PPV values for different risk groups are computed. Can this procedure be broken down into simpler steps? My current understanding is that the initial cutoff between low-risk and not low-risk is determined by the Youden index, which is essentially some point on the ROC curve. The associated threshold/cutoff value of the point on the ROC curve partitions the subjects into two groups – low-risk and not low-risk. The latter is partitioned into three equally sized groups based off their probability. Then, an additional confusion matrix for original cancer/non-cancer is computed for each of these groups and the PPV is reported as their respective cutoffs. This makes sense because one doesn’t want a high number of false positives if a patient is low-risk but would increasing tolerate a high number of false positives the higher risk a patient is. If my understanding is correct, what is the PPV of the low-risk group?
- What are the PPVs for the test set?
- In Table 2, it is okay to report sensitivity and specificity of the testing set as is. But, accuracy is biased by the imbalanced nature of the test set, as is AUROC. Consider a weighted accuracy and precision-recall curve instead, respectively. This also help with comparison to the training performance, which is based on balanced classes.
- How do the descriptive statistics change for the top 3 cancers as well as the distribution of cancer/non-cancer classes?
- "When the top 3 most possible organ system were reported" – does this mean that the read-world clinical data reports more than one possible organ system? How were the subjects labeled then? The method makes sense, but this aspect of the reporting of the results does not. Or perhaps is that you only cross-validated the knn model on subjects who had cancers in the top 3 most common categories? Please add some clarification. Or is that 3 neighbors are used, and this was the best balanced between sensitivity/specificity? (aside: after reading the discussion, it seems this algorithm recommend N organ systems, and the statistics are computed based on whether the true organ system is included in this top 3).
- I believe Figure 2 is missing.
- Consider using chi-squared test to characterize the correlation between risk scores and stage.
- The authors provide no evidence that they have developed and validated ML-derived software that "substantially improves tumor biomarker testing". No results are references are reported for single biomarkers or current clinical practices. Please revise lines 278-279.
- Figure 4 reports (mean?) TTD while another part of the manuscript reports median. Why is this?
- The results are not elaborated on. Here are a few suggestions. Why do certain methods outperform other methods? On lines 87-88, the authors state that ML outperforms traditional methods, but logistic regression outperformed ML method for males. Why does there seem to be generalization of the models for males but not for females?
- Have the authors considered ensembling all three models?
Author Response

(The authors gave the same response as above.)

Round 2
Reviewer 1 Report
All the raised issues were reasonably met.
Reviewer 2 Report
The authors have addressed all my concerns.